# Advances and future trends in research on carbon emissions reduction in China from the perspective of bibliometrics

Caiyun Chen[1,2], Wei Liu[2]*

1 Party School of Nanjing Municipal Committee of CPC, Nanjing, China, 2 Key Laboratory of Poyang Lake Environment and Resource Utilization, Ministry of Education, School of Resources and Environmental, Nanchang University, Nanchang, China

* LiuW@ncu.edu.cn

## Abstract

Addressing global warming is one of the most pressing environmental challenges and a crucial agenda for humanity. In this literature study, we employed bibliometrics to reproduce nearly two decades of research on carbon emission reduction in China, the largest carbon emitter worldwide. The scientometrics analysis was conducted on 1570 academic works published between 2001 and 2021 concerning China's carbon emission reduction to characterize the knowledge landscape. Using CiteSpace and VOSviewer, the basic characteristics, research forces, knowledge base, research topic evolution, and research hotspots were identified and revealed. The analysis results show that the attention to and research on China's carbon emissions have increased in recent years, giving rise to leading institutions and relatively stable core journal groups in this field. The research disciplines are relatively concentrated, but the research collaboration needs strengthening. The research hotspots are mainly carbon emission causes, impacts, and countermeasures in China, and the research frontiers have been constantly advanced and expanded. In the future, research on countermeasures needs more effort, and research cooperation needs to strengthen. The changing landscape of hotspot clusters reveals China's transition towards a low-carbon economy. Through comprehensive analysis of the potential and obstacles to China's transition to low-carbon development, we identified three promising areas of action (low-carbon cities, low-carbon technologies and industries, and transforming China's energy system) and proposed research directions to address remaining gaps systematically.

## 1. Introduction

Global warming is one of the most pressing environmental challenges worldwide [1–4], and an economic, social, and political issue reported to worsen. If global warming continues at the current rate, the global average temperature may rise by 1.5°C by 2030–2052 [5–7]. Although global carbon emissions dropped unprecedentedly by 5.4% in 2020, they are rebounding rapidly. The emission gap remains large, and the net-zero emission goal is far ahead [8]. In response to global warming, the international community, countries, cities, and non-

**Data Availability Statement:** The underlying dataset of this study is publicly available from the Dryad repository (https://doi.org/10.5061/dryad.jwstqjqg0). Considering that the DOI retrieval link is not activated, we are providing an alternative way

to access the data: (1) We have generated a URL for data sharing: https://datadryad.org/stash/share/BucNXP-pYp8uiqE-nCnfaYfUC__9QxRMLW4W0d0n9MI. (2) We are providing a data download link along with login information. Data Link: https://datadryad.org/stash/dataset/doi:10.5061/dryad.jwstqjqg0. Account: 0009-0005-1767-5295; Password: Liuwei003.

**Funding:** The authors received no specific funding for this work.

**Competing interests:** The authors have declared that no competing interests exist.

governmental organizations have taken various initiatives, such as the United Nations Framework Convention on Climate Change (UNFCCC; 1992), Kyoto Protocol (1997), and Paris Agreement (2015) [9–12].

China is the most populous country and the second-largest economy globally. With economic growth and energy consumption, China's carbon emissions are becoming an increasingly prominent problem. Statistics show China overtook the US as the largest carbon emitter globally around 2004. In 2021, China's carbon emissions exceeded 11.9 Gt, accounting for about 33% of global emissions. The carbon emissions from China's international trade have been significantly scrutinized [13–17]. These net emissions mainly originate from exports to Europe, the US, Japan, and South Korea [18]. Excluding the emissions embodied in exports, China remains the second-largest carbon emitter globally in terms of consumption-based emissions, following only the US [19]. Previous studies have examined China's nationwide consumption-based emissions by considering it a homogeneous national entity [16, 19] and found its consumption-based emissions consistently lower than production-based emissions over the past decade.

China's carbon emissions have become an increasingly hot issue in the international community. China has pledged to become carbon neutral by 2060 and has employed various strategies to minimize carbon emissions [20]. To better understand the research status, evolution law, and development trend in this field, scholars have adopted bibliometrics to analyze and summarize the issues related to China's carbon emissions and achieved rich research results. Extensive research has been conducted on China's carbon emission trading [21–24]. Empirical research has been carried out on carbon emissions in Chinese provinces [25–28], such as Xinjiang, Hunan, and Inner Mongolia, and Chinese cities, such as Shanghai, Wuhan, and Xi'an [29–31]. Factors influencing carbon emissions have been investigated, such as energy structure change, population flow, and higher education development [32–34].

However, most bibliometrics studies on China's carbon emission reduction are limited to a certain thematic area [21–28]. The advances and future trends in the research on China's carbon emission reduction have not been studied based on bibliometrics. Through an in-depth analysis of the development context and trends of research on China's carbon emissions, this paper aims to present the current situation and concerns of China's carbon emission reduction work and promote the realization of China's carbon emission reduction goals, especially the "3060" goal.

Therefore, we combined systematic mapping and bibliometrics to perform a global scientometric analysis of China's carbon emissions research. The collected papers can promptly reflect the trends at the frontiers of science to some extent, and statistical analysis and content mining are performed through bibliometrics [35, 36]. The primary goal is to systematically map the evolution of China's carbon emission research over time and identify the evolution of key research themes using a network of co-cited references and a network of co-occurring keywords. The secondary goal is to provide measures of research networks (countries, institutions, authors, and journals) and detect research richness, gaps, emerging trends, biases, and limitations. Finally, we comprehensively analyzed the potential and obstacles to China's transition to low-carbon development, identifying three promising areas of action (low-carbon cities, low-carbon technologies and industries, and the transformation of China's energy system), and proposed research directions to address remaining gaps systematically.

## 2. Methods

### 2.1 Search strategy and data collection

We adopted text mining and visual analysis software in this bibliometrics study. CiteSpace, Gephi, VOSviewer, and other information visualization software were used to analyze the

literature on China's carbon emissions retrieved from the WoS Core Collection database on June 22$^{nd}$, 2022. We searched the WoS Core Collection database for works published from 2001 to 2021, with "China" and "carbon emissions" in their titles. The document type was limited to "Article", and the Science Citation Index-Expanded (SCI-E) and Social Sciences Citation Index (SSCI) were selected. We obtained 1570 articles, including papers, editorial materials, letters, news, and conference abstracts. Since papers more reliably reflect research trends, we only considered the data related to research papers (1504, 95.75%) for the corresponding analysis to improve the validity of bibliometrics.

## 2.2 Analysis methods

Bibliometrics is a quantitative analysis method to study the external characteristics of documents. It uses statistical methods to describe, analyze, evaluate, and predict the development status and trends of research objects. Bibliometrics has been widely used in analyzing disciplinary structures and development trends [37]. Knowledge maps are an information visualization tool to systematically and comprehensively analyze the research status, hotspots, and future trends of specific research fields based on bibliometrics [38–40]. This intuitive, efficient, and convenient tool has the dual nature of "graph" and "spectrum", combining various data mining and statistical methods. Since being proposed in the US in 2003, it has developed rapidly into an important method for disciplinary research. Scholars and enterprises have developed software for drawing scientific knowledge maps, and the most commonly used among them is CiteSpace [41], developed by Chen et al. of Drexel University and can be used to draw cooperation maps, co-occurrence maps, and co-citation maps [42].

Based on the bibliometrics method, this study used the Derwent Data Analyzer for text mining and CiteSpace, Gephi, and VOSviewer for visual analysis, which are Java applications for visualizing and analyzing trends and patterns in scientific literature. Designed as a tool for progressive knowledge domain visualization, it focuses on finding critical points in the development of a field or a domain, especially intellectual turning points and pivotal points [40]. We quantitatively analyzed the literature on China's carbon emissions from the WoS Core Collection database and visualized the results. The findings revealed the current status and trends of research on China's carbon emissions in terms of publishing trends, publishing institutions, core authors, research institutions, research topics, and research hotspots. The study facilitates the accurate grasping of the development situation and evolution trends of research on China's carbon emissions, promotes subsequent research, and provides references for carbon emission research in other countries and regions worldwide.

We proposed a comprehensive analysis framework to interpret the trends and thematic changes of research on China's carbon emissions in 1504 papers published from 2001 to 2021. Specifically, one part of this study analyzed the current research, including annual publication volume, core research institutions, and high-yield journals/researchers, to understand the overall research progress. The other part analyzed the research hotspots, including highly cited papers, keyword co-occurrence networks, and burst detection, to summarize the research focus and identify possible research trends in the future.

## 3. Results

### 3.1 Analysis of current research

**3.1.1 Annual publication volume.** The annual publication volume and its trends are important for assessing the current status and research trends. Fig 1 shows a generally steady growth in papers on China's carbon emissions from 2001 to 2021. According to the trends of publication volume, the Chinese carbon emission research can be divided into the start-up

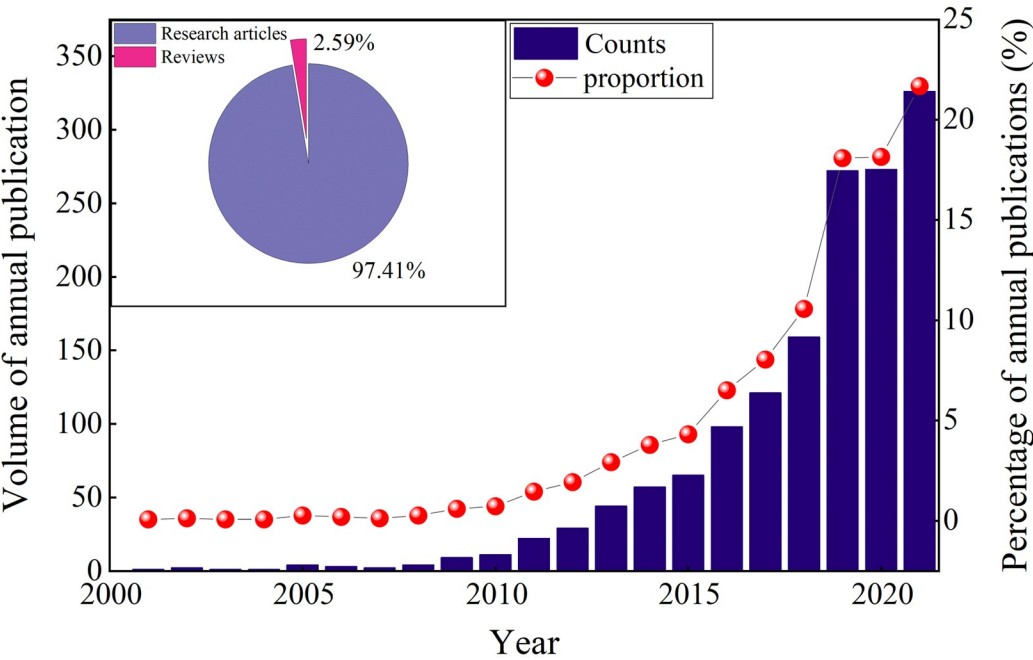

**Fig 1. The trend of papers published from 2001 to 2021.**

period (2001–2009), development period (2010–2015), and outbreak period (2016–2021). During the start-up period, less than ten relevant articles were published per year, and China's economy and carbon emissions grew rapidly. In 2007, the Chinese government explicitly proposed to "develop a low-carbon economy" for the first time. Accordingly, the number of related papers in China grew steadily. However, very few papers were published internationally, indicating that China's carbon emissions failed to attract enough international attention. Since 2011, the number of papers published has grown rapidly. As the "Beautiful China" concept proposed at the 18th National Congress of the Communist Party of China in 2012, ecological civilization construction has been emphasized, and China's environmental protection has been increasingly valued by the government, enterprises, and academia. From 2010 to 2015, the annual publication volume increased nearly 7 times from 11 to 75. In 2015, the Chinese government proposed a vision of innovative, coordinated, green, open, and inclusive development, i.e., the Five Development Concepts". Green development means protecting and restoring the ecological environment and developing a circular economy and low-carbon technologies to promote the harmonious development of the economy, society, and nature. The importance of ecological conservation has been raised to an unprecedented level. Against this background, the number of papers on carbon emissions in China has grown exponentially, and new theories, perspectives, methods, and technologies have emerged constantly. The total number of research achievements in 2021 is over three times that in 2016 and nearly five times that in 2015.

**3.1.2 Core research institutions.** Analyzing research institutions helps to understand the distribution and research strength of important research teams in the field and facilitates the tracking of follow-up research. Based on statistics from the WoS Core Collection database, we used the Derwent Data Analyzer as the text mining tool to obtain the top 10 core research institutions (Fig 2), including the Chinese Academy of Sciences, Tsinghua University, North China Electric Power University, University of Chinese Academy of Sciences, Beijing Institute

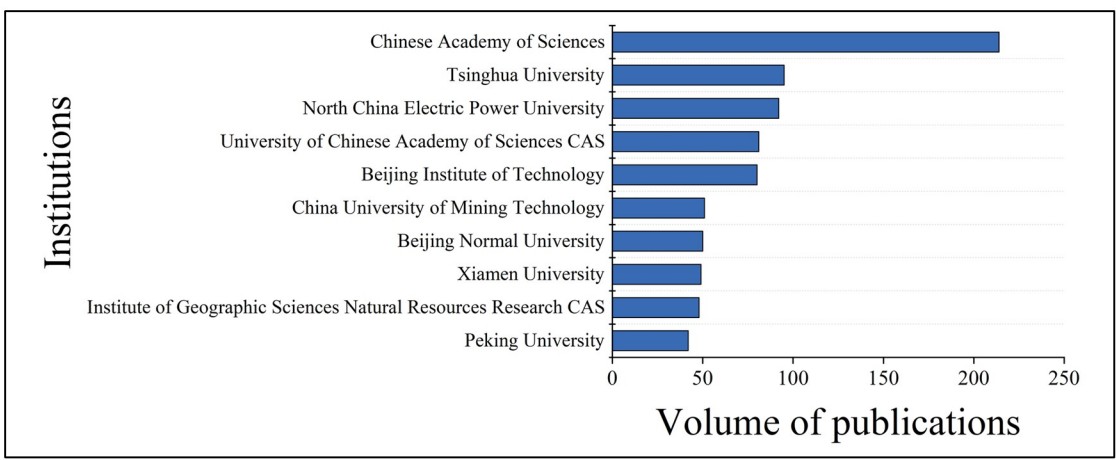

**Fig 2. Core research institutions.**

of Technology, Xiamen University, China University of Mining Technology, Institute of Geographic Sciences Natural Resources Research CAS, Beijing Normal University, and Peking University. These institutions published 833 papers, accounting for 55.39% of the papers analyzed. The top-ranking Chinese Academy of Sciences published 222 papers, accounting for 14.76% of the papers analyzed. These findings suggest that all the core research institutions except the Chinese Academy of Sciences, a scientific research institution, are national universities, highlighting the core position of national universities in this research field. However, research on China's carbon emissions involving the energy revolution, technological revolution, and industrial change requires the active participation of the whole society. Large enterprises, especially big energy companies, have played a vital role in the research on Chinese carbon emissions but fell very short of research institutes, according to statistics. In addition, empirical studies were the focus and trend of research. However, more research institutions are needed due to apparent differences in China's regional industrial structures, economic development, and environmental protection policies. In addition to national universities, enterprises, local universities, think tanks, and social institutions should jointly promote research on China's carbon emissions.

**3.1.3 Institutional cooperation network.** Fig 3 shows that the institutional cooperation network has 133 nodes and 131 connections, with a network density of 0.0149, indicating room to increase the overall connectivity of the cooperation network for research on China's carbon emissions. The connections between teams are relatively scattered, and knowledge sharing and interaction need to be improved. In addition, much room remains for enhancing research cooperation in this field. The Chinese Academy of Sciences, Tsinghua University, Beijing Institute of Technology, and the University of Chinese Academy of Sciences cooperated relatively actively. In particular, the Chinese Academy of Sciences has a cooperation frequency of 196. The Chinese Academy of Sciences was the first institution to cooperate in 2005. The Beijing Institute of Technology, Xiamen University, Beijing Normal University, and various research institutions started cooperation relatively early. From the perspective of centrality, the Beijing Institute of Technology (0.78), the Chinese Academy of Sciences (0.69), Tsinghua University (0.48), and the China University of Geosciences (0.31) were important bridges for cooperation.

**3.1.4 High-yield journals.** Statistics on the distribution of published journals help understand the subject distribution and research dynamics in a specific field. According to statistics

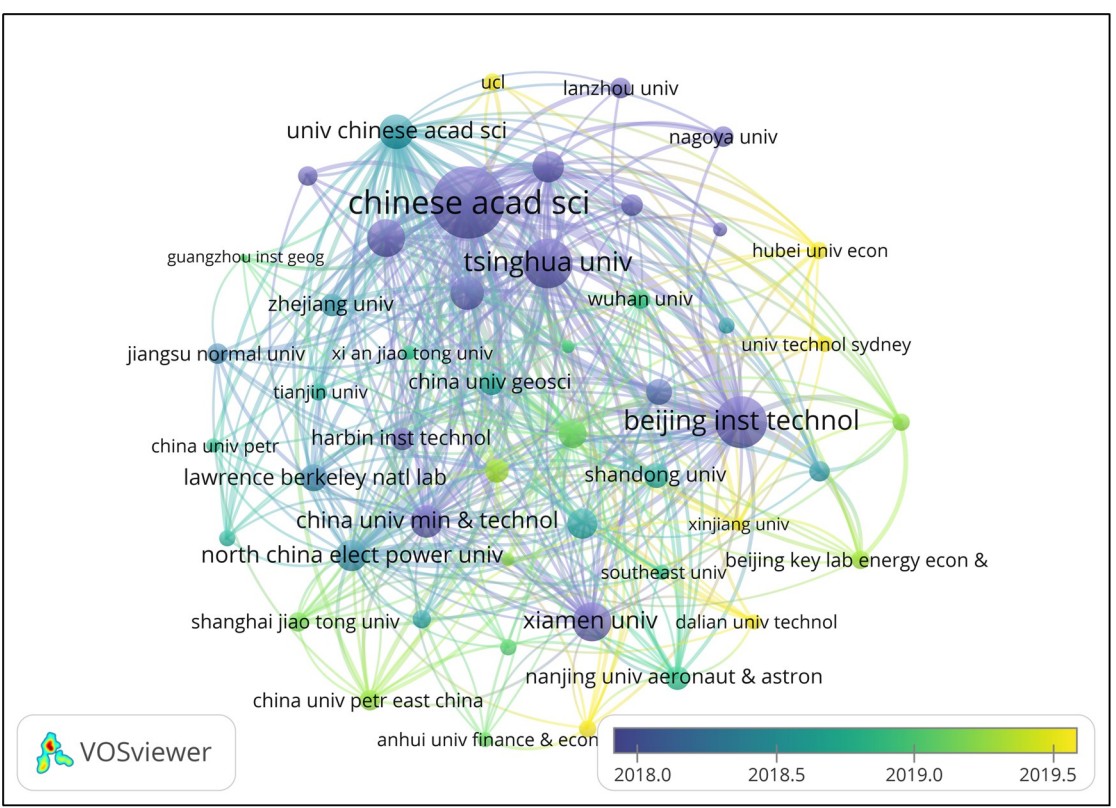

**Fig 3. Institutional cooperation network.**

on journals publishing articles on China's carbon emissions in the WoS Core Collection database, we obtained the top 10 journals by publication volume (Table 1). The top 3 journals are the *Journal of Cleaner Production*, *Sustainability*, and *Environmental Science and Pollution Research*, which are essential knowledge carriers and dissemination platforms for research on China's carbon emissions. Among them, the top-ranking *Journal of Cleaner Production* has 194 publications accounting for 12.90% of the total sample, indicating that it is vital in research

**Table 1. Top 10 journals in carbon emission research.**

| No. | Journal | TP | Percentage | IF |
|---|---|---|---|---|
| 1 | *Journal of Cleaner Production* | 194 | 12.93% | 11.072 |
| 2 | *Sustainability* | 129 | 8.58% | 3.889 |
| 3 | *Environmental Science and Pollution Research* | 93 | 6.19% | 5.190 |
| 4 | *Energy Policy* | 73 | 4.86% | 7.576 |
| 5 | *Science of the Total Environment* | 62 | 4.12% | 10.753 |
| 6 | *Applied energy* | 48 | 3.19% | 11.446 |
| 7 | *Energy Economics* | 44 | 2.92% | 9.252 |
| 8 | *International Journal of Environmental Research and Public Health* | 41 | 2.72% | 3.390 |
| 9 | *Energies* | 37 | 2.46% | 3.252 |
| 10 | *Energy* | 37 | 2.46% | 8.857 |

**Note**: TP = the number of publications, IF = impact factor.

on China's carbon emissions. The second-ranking *Sustainability* has 129 publications accounting for 8.58% of the total sample. The top 10 journals published 758 papers, accounting for 50.40% of the total sample. These results show that the journals related to research on China's carbon emissions have a high clustering degree, gradually forming a relatively stable group. The journals are mainly in the categories of environmental science (877), green sustainable science technology (397), environmental studies (304), energy fuels (261), engineering environment (257), and economics (203). In terms of research orientation, the leading journals mainly fall into the fields of environmental science, ecology, science and technology, engineering, energy fuels, and business economics. The trend of intersectionality is apparent. Despite the prominent position of carbon emission research in China, none of the core journals originate from China, indicating the lack of international influence of Chinese publications. Therefore, China's core publications should be internationalized and integrated with international standards to gain a higher international status. Moreover, the current core journals are mainly in the environment, energy, and other disciplines. However, carbon emissions are comprehensive, involving economic, political, technological, social, and other fields. Therefore, the research field needs expansion and enrichment, and frontier research across fields needs attention.

## 3.2 Research hotspots

**3.2.1 Highly cited papers.** Essential Science Indicators (ESI) is an analytical tool to identify top-performing research in the WoS Core Collection database. With citation frequency ranking 1% in the same academic field, ESI highly cited papers can objectively reflect the research trends and hotspots. Table 2 lists the top 10 highly cited papers on China's carbon emissions published from 2009 to 2017, 4 of which were published in 2014. The most cited paper, titled *Energy Consumption, Carbon Emissions, and Economic Growth in China*, was published in 2009 [43]. Its corresponding author is Zhang from the School of Business & Administration, North China Electric Power University. That paper investigated and analyzed the logical connection between economic growth, energy consumption, and carbon emissions in China through empirical methods, concluding that neither carbon emissions nor energy consumption would lead to China's economic growth. Based on this research, Zhang and Cheng suggested that the Chinese government adopt energy conservation and carbon emission reduction policies in the long run without hindering economic growth. Another critical article was *Reduced Carbon Emission Estimates from Fossil Fuel Combustion and Cement Production in China* [44] published on *Nature* in 2015 (Fig 4), authored by over 20 academic researchers, including Zhu Liu from the John F. Kennedy School of Government, Harvard University. In that paper, the authors performed targeted research on carbon emissions from fossil fuel combustion and cement production in China, using updated harmonized data on energy consumption and clinker production and two new comprehensive sets of measured emission factors, and finally revised the existing conclusions. The overall analysis of the highly cited papers shows that, in recent years, the focus of research on China's carbon emissions mainly includes the logical connection between carbon emissions, energy consumption, and economic growth; influencing factors and countermeasures of China's carbon emissions; and empirical research on China's carbon emissions.

**3.2.2 Keyword co-occurrence network.** Keywords reflect the research theme to a certain extent. Keyword co-occurrence networks can clearly show the high-frequency keywords in literature and the co-occurrence relationship between keywords. As shown in Fig 5, a close network map means a strong correlation between keywords. The large number of keywords indicates relatively extensive research on China's carbon emissions. Frequency analysis of

**Table 2. Top 10 highly cited papers published from 2001 to 2021.**

| No. | First Author | Article | Journal | year | citation |
|---|---|---|---|---|---|
| 1 | Xing-Ping Zhang | Energy Consumption, Carbon Emissions, and Economic Growth in China | Ecological Economics | 2009 | 818 |
| 2 | Zhu Liu | Reduced Carbon Emission Estimates from Fossil Fuel Combustion and Cement Production in China | Nature | 2015 | 758 |
| 3 | Shobhakar Dhakal | Urban Energy Use and Carbon Emissions from Cities in China and Policy Implications | Energy Policy | 2009 | 527 |
| 4 | Ching-Chih Chang | A Multivariate Causality Test of Carbon Dioxide Emissions, Energy Consumption and Economic Growth in China | Applied Energy | 2010 | 347 |
| 5 | Afeng Zhang | Effect of Biochar Amendment on Maize Yield and Greenhouse Gas Emissions from a Soil Organic Carbon Poor Calcareous Loamy Soil from Central China Plain | Plant Soil | 2012 | 308 |
| 6 | Shi-Chun Xu | Factors that Influence Carbon Emissions due to Energy Consumption in China: Decomposition Analysis Using LMDI | Applied Energy | 2014 | 299 |
| 7 | Yue-Jun Zhang | Can Environmental Innovation Facilitate Carbon Emissions Reduction? Evidence from China | Energy Policy | 2017 | 298 |
| 8 | Ke Wang | China's Regional Industrial Energy Efficiency and Carbon Emissions Abatement Costs | Applied Energy | 2014 | 286 |
| 9 | Shao-jian Wang | Urbanisation, Energy Consumption, and Carbon Dioxide Emissions in China: A Panel Data Analysis of China's Provinces | Applied Energy | 2014 | 280 |
| 10 | Yue-Jun Zhang | The Impact of Economic Growth, Industrial Structure and Urbanization on Carbon Emission Intensity in China | Nat Hazards | 2014 | 264 |

relevant keywords suggests that in research on China's carbon emissions, carbon emissions are closely related to energy consumption, economic development, and climate change. Many key nodes have formed in research on China's carbon emissions. The keywords with the highest centrality are energy consumption, economic growth, climate change, and urbanization,

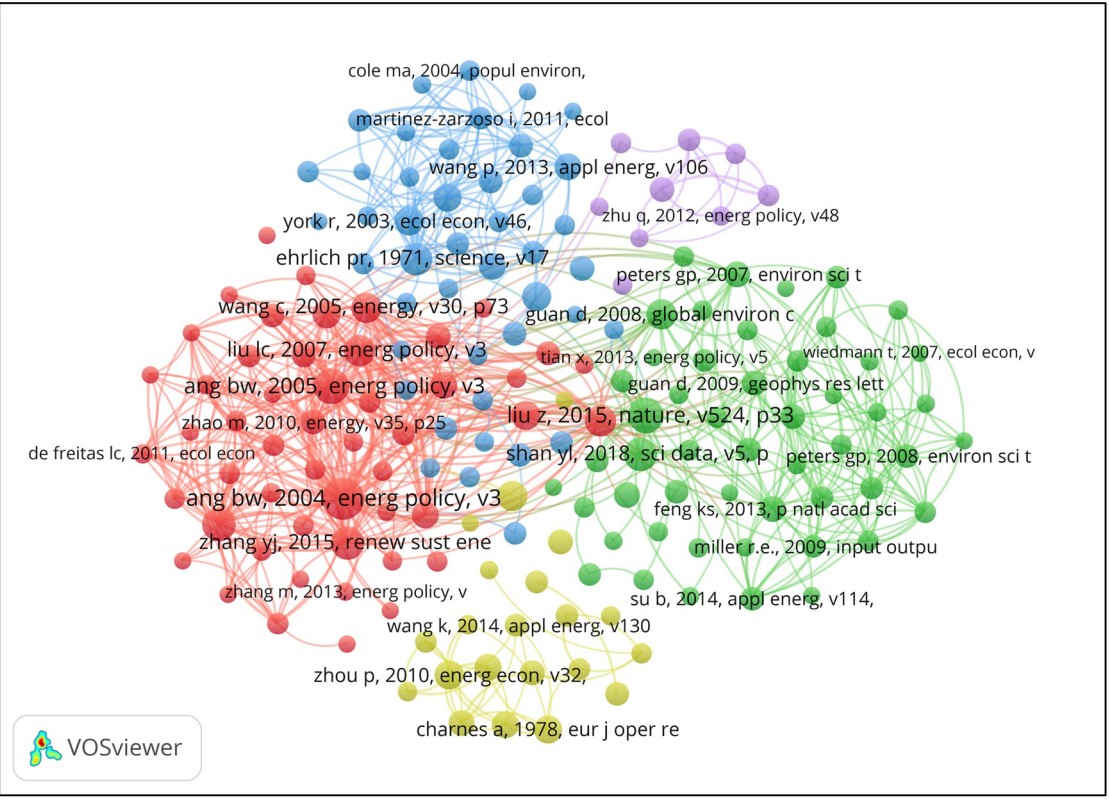

**Fig 4. Co-citation references network (2001–2021) obtained with VOSviewer.**

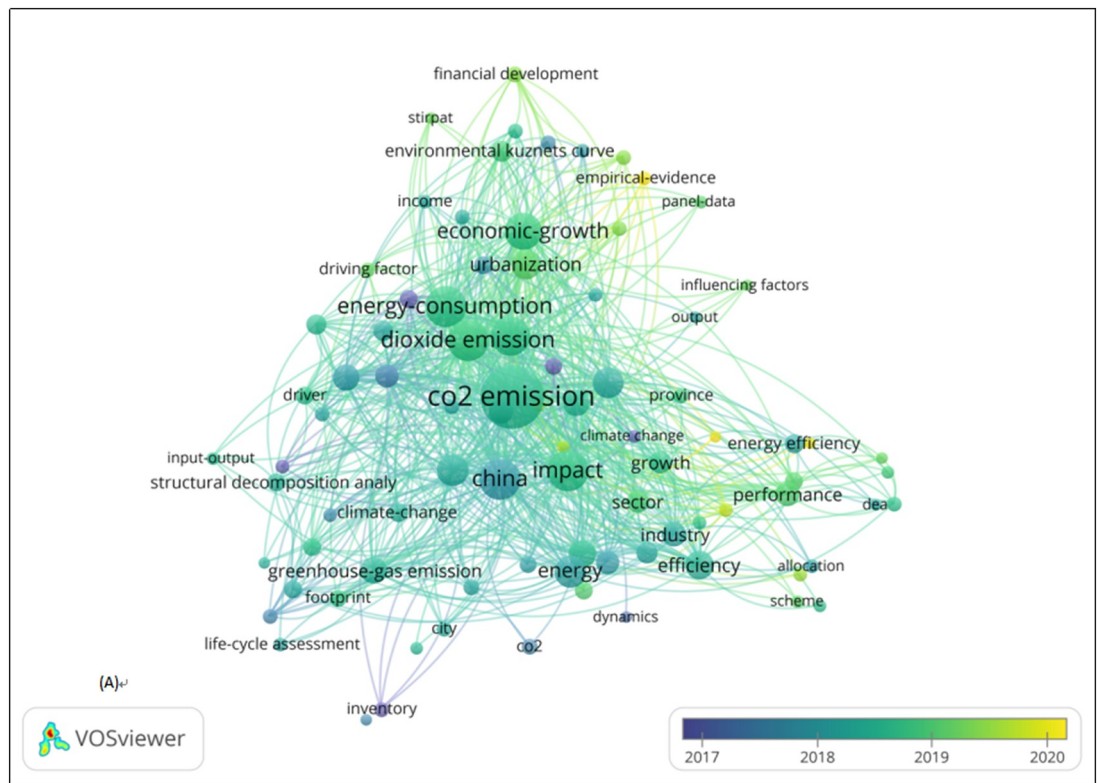

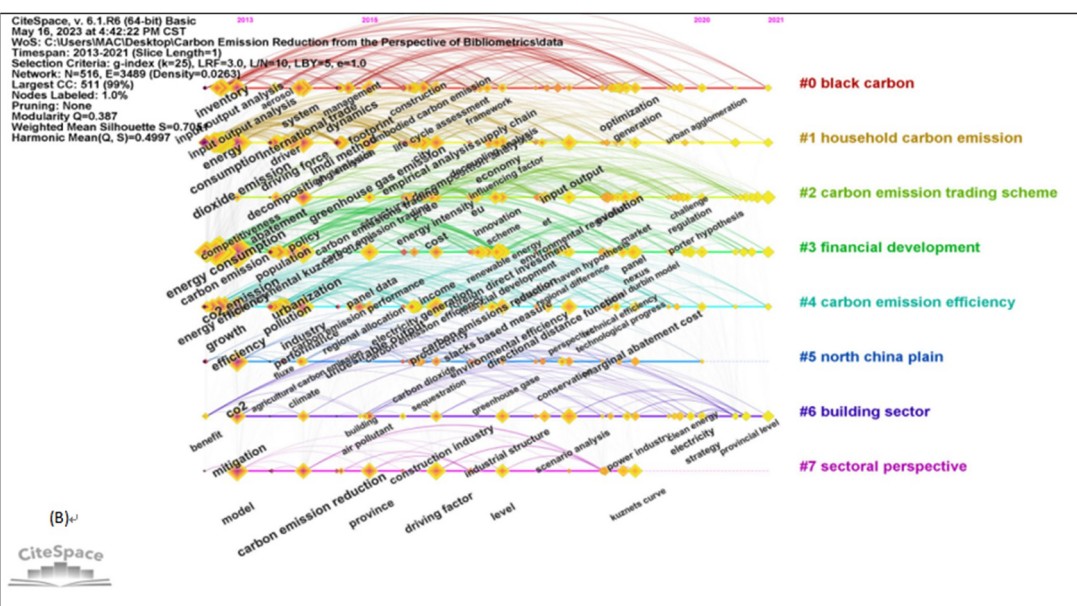

**Fig 5. (A) Keyword co-occurrence network and (B) Timeline visualization of co-occurring keywords networks.**

which constitute the key path of the domain knowledge network of China's carbon emissions. In terms of evolution trends, many keyword co-occurrence networks in the early days formed with factors affecting carbon emissions as the focus, such as energy consumption. With research development, keyword co-occurrence networks have formed from analyses of various models, technologies, industries, and markets (Fig 5), constituting the key path of China's carbon emissions domain knowledge network. Furthermore, studies on policies, systems, and management of China's carbon emissions are significant. However, their keyword co-occurrence networks have not been formed. Additionally, relevant social economics, such as green finance, need further attention.

### 3.2.3 High-frequency keywords.

To a certain extent, high-frequency keywords reflect the research hotspots in a certain field over a certain period. Fig 6 shows the keyword density map related to carbon emission, and the top 15 high-frequency keywords are presented in Table 3. The self-directed keywords, such as China and carbon emissions, have been removed. The top 15 high-frequency keywords indicate that the research is relatively straightforward in scope and specific in content, mostly involving influencing factors such as energy consumption, climate change, economic growth, and urbanization. In addition, the research highlights input-output analysis, LMDI, STIRPAT model, scenario analysis, carbon emission efficiency analytical methods, tools, and models. As one of the main influencing factors on carbon emissions, energy consumption has been studied relatively centrally. Firstly, energy consumption, carbon emissions, and economic growth are closely related. Based on various models, analytical tools, and specific data, the logical connection between the three has been analyzed to make recommendations for China's energy structures, industrial economy, and carbon emission reduction

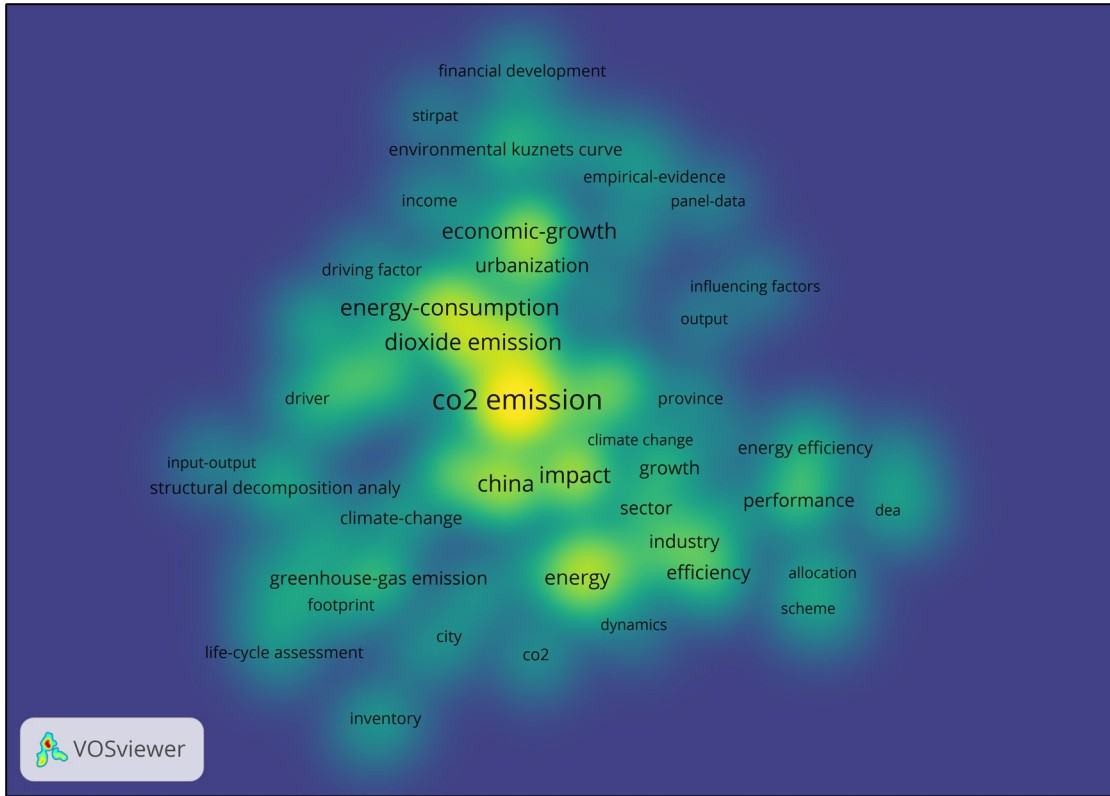

**Fig 6. Keyword density map related to carbon emission.**

**Table 3. Top 15 high-frequency keywords.**

| No. | Keyword | Frequency | Links | Total link strength |
|---|---|---|---|---|
| 1 | Energy consumption | 63 | 32 | 101 |
| 2 | Input-output analysis | 42 | 18 | 52 |
| 3 | Economic growth | 39 | 27 | 68 |
| 4 | Urbanization | 38 | 20 | 56 |
| 5 | LMDI | 37 | 24 | 56 |
| 6 | Climate change | 33 | 20 | 38 |
| 7 | Structural decomposition analysis | 30 | 14 | 39 |
| 8 | Scenario analysis | 28 | 17 | 30 |
| 9 | Influencing factors | 25 | 12 | 28 |
| 10 | Carbon emission efficiency | 24 | 9 | 14 |
| 11 | Data envelopment analysis | 24 | 14 | 20 |
| 12 | Embodied carbon emissions | 24 | 9 | 19 |
| 13 | STIRPAT model | 24 | 15 | 36 |
| 14 | Carbon emission intensity | 21 | 21 | 28 |
| 15 | Carbon emissions trading | 21 | 6 | 12 |

*In the table, "links" indicate the degrees of co-occurrences between keywords and other words.

policies [45, 46]. Therefore, these issues have become research hotspots relatively early. Secondly, with the continuous progress of research methods and tools and the deepening and concretization of research, the quantitative and empirical analyses on energy consumption and carbon emissions have been enriched, including spatial econometric analysis, driving factor analysis, and empirical analysis on provinces and cities [47–51]. Furthermore, more attention is paid to the influencing factors on energy consumption and carbon emissions, such as urbanization, new energy consumption, urban residences, and household consumption [52, 53].

**3.2.4 Burst detection.** Burst words are keywords that suddenly increase in number within a certain period, and burst detection can effectively reveal the active directions or themes in the development of the discipline, i.e., the research frontier. Table 4 lists the top 20 keywords in the research on China's carbon emissions. The keyword with the highest emergence intensity (5.05) is "carbon emission efficiency", indicating that the research on carbon emission efficiency has attracted extensive attention from academia [54, 55]. The keyword with the longest emergence time is "greenhouse gas (GHG)", an active theme from 2005 to 2013 [56, 57]. According to burst detection, the frontier development in China's carbon emissions can be divided into three stages. (1) Many emergent words have appeared since 2005, including GHG [56], emission inventory [58], and international trade [59], suggesting that research on China's carbon emissions has gradually attracted scholarly attention. Although some accomplishments were achieved in certain domains, the research was still in its initial and exploratory stages. The research contents and methods were relatively superficial and single, and further in-depth and specific discussions were required [56]. (2) Since 2009, research on China's carbon emissions has developed rapidly, with many keywords emerging, such as the computable general equilibrium model [60], energy efficiency [61], shadow price [61], and the STIRPAT model [62]. Practical applications of various analytical models and tools enriched the research methodologies, showing increasing attention to empirical research. (3) Since 2019, more and more keywords have emerged, such as carbon emission trading, spillover effect, construction industry, carbon intensity, carbon emission peak, and regional difference, highlighting the new

**Table 4. Top 20 Keywords with the strongest citation bursts.**

| Keywords | Strength | Begin | End | 2001–2021 |
|---|---|---|---|---|
| Greenhouse gas | 2.18 | 2005 | 2013 | |
| Emission inventory | 2.89 | 2006 | 2013 | |
| International trade | 2.14 | 2008 | 2014 | |
| Computable general equilibrium model | 2.13 | 2009 | 2014 | |
| Energy efficiency | 1.92 | 2013 | 2014 | |
| Shadow price | 1.82 | 2014 | 2015 | |
| Stirpat model | 1.98 | 2017 | 2019 | |
| Carbon emission efficiency | 5.05 | 2019 | 2021 | |
| Spatial Durbin model | 4.28 | 2019 | 2021 | |
| Carbon emission trading | 3.62 | 2019 | 2021 | |
| Spillover effect | 2.85 | 2019 | 2021 | |
| Paris Agreement | 2.84 | 2019 | 2021 | |
| Construction industry | 2.61 | 2019 | 2021 | |
| Carbon intensity | 2.36 | 2019 | 2021 | |
| Regional difference | 2.17 | 2019 | 2021 | |
| Decoupling analysis | 2.14 | 2019 | 2021 | |
| Directional distance function | 2.12 | 2019 | 2021 | |
| Structural decomposition analysis | 1.99 | 2019 | 2021 | |
| Carbon emission intensity | 1.86 | 2019 | 2021 | |
| Input-output analysis | 1.84 | 2019 | 2021 | |

perspectives, new fields, and methodologies of research on China's carbon emissions, involving the causes, impacts, tendencies, responses, and regional comparisons of China's carbon emissions [63–65]. The emerging research has certain stability and continuity, and the above keywords continue to 2021, representing the frontier progress and development trends of current academic research.

## 4. Discussion

### 4.1 Efforts and bottlenecks of China's dual carbon goal implementation

**4.1.1 Exploring paths to building low-carbon cities.** As the world's largest developing country, China is undergoing an unprecedented scale of urbanization, with the urbanization rate growing from around 36% in 2000 to about 53% in 2012 [66]. This trend involves migration, urban expansion, and new cities emerging near existing ones, thus leading to persistent and severe stress on infrastructure, economic growth, land development, urban resource demands, and pollution [67]. The 35 largest Chinese cities are home to approximately 18% of the country's population and are responsible for 40% of its energy consumption and $CO_2$ emissions [68, 69]. As China's urbanization continues, this percentage is expected to grow even higher. Therefore, implementing low-carbon urban development strategies holds great potential for China to reduce GHG emissions.

To achieve the goals of sustainable urbanization, ecological civilization, and scientific development, the PRC Ministry of Housing and Urban-Rural Development and the Worldwide Fund for Nature launched the initial low-carbon city program in 2008, followed by China's National Development and Reform Commission (NDRC) in August of 2010, initiating a low-carbon city experiment in eight cities, including Tianjin, Chongqing, Shenzhen, Xiamen, Hangzhou, Nanchang, Guiyang, and Baoding [70]. The concept of low-carbon cities combined

with elements of the low-carbon economy and society provides a new approach to China's sustainable development [68, 71]. These low-carbon cities aim to promote low-carbon lifestyles and develop low-carbon economies with the defining characteristics of reduced pollution, lower emissions, and higher energy efficiency [72].

Decoupling research exploring the relationship between economic level and GHG emission has gained attention. Some of China's large cities have experienced strong decoupling effects between economic development and GHG emissions. However, given the significant economic imbalances among Chinese cities, it is challenging to generalize these findings. While adjusting industrial structure is often suggested, the increased GHG emissions in many cities are attributed to the tertiary sector rather than the industrial sector. Researching emissions from the tertiary and household sector is a promising strategy for GHG emission control. Energy systems and energy intensity are crucial to emissions. Although city layout is critical, detailed academic discussion in this area is lacking, and improving city designs is particularly challenging for existing cities [73]. The concept of low-carbon cities has been embraced worldwide as cities are significant drivers of energy consumption and associated carbon emissions. By integrating elements of a low-carbon economy and society, low-carbon cities offer a new model for China to achieve sustainable urbanization, emphasizing ecological civilization and scientific development [74]. Governmental understanding of the most important features of low-carbon cities is only the first step; the most challenging issue is transitioning from a traditional development path to a low-carbon one. Zhang et al. employed the Long-range Energy Alternatives Planning (LEAP) model to analyze the pathways necessary to transform Beijing's economic and social development pattern to achieve low-carbon energy consumption and low carbon emission from 2007 to 2030 [74]. They propose clean energy policies to accelerate city energy structure changes necessary to mitigate GHG emissions. Similarly, Guan and Barker conducted a scenario analysis to identify interdependent technological improvements and production structure changes key to determining factors affecting both carbon intensity and carbon emissions in Guangyuan [75]. Their findings suggested that government policies should emphasize decarbonization of the production structure to prevent the mistake of "polluting first and addressing the pollution later." Low-carbon city development aims to combine cleaner and ecologically sustainable economic growth with developing knowledge cities by bringing in knowledge, expertise, and technology from developed countries. Some researchers considered emulating successful low-carbon cities in developed countries useful for China's cities to achieve low-carbon development. In addition, de Jong et al. constructed a typology of Sino-foreign initiatives focusing on ecological knowledge related to cities to identify conditions for robust Sino-foreign partnerships in eco-cities [76].

Studies show that Chinese cities, like those worldwide, confront the dilemma of balancing economic growth and decarbonization. As such, researchers are exploring ways to coordinate industry and eco-efficiency to implement low-carbon city strategies. Some researchers proposed industrial symbiosis (IS) as a viable option for promoting sustainable urban development by achieving system-level innovation. IS is an innovation that creates value, reduces costs, and improves the environment by sharing service, utility, and by-product resources across diverse industrial processes or actors [73, 77].

**4.1.2 Low-carbon technologies and industries.** Developing low-carbon emission industries is crucial for promoting low-carbon cities. Industrial production is the pillar of China's economy and generates a large portion of the country's emissions. Policies to reduce energy consumption and carbon emissions often target energy-intensive industrial sectors, such as iron and steel, nonferrous metals, petrochemicals, chemicals, machinery, and textiles [73]. In 2013, the PRC Ministry of Industry and Information Technology set the goal to reduce carbon emissions per unit of industrial added value by 17% to 22% by 2015. Improving energy

efficiency is crucial for low-carbon industrial development. China is building carbon emission trading systems (ETS) targeting energy-intensive industrial sectors [70, 73]. As a developing country, China is pressured to maintain economic growth and reduce carbon emissions. Achieving a balance between these two goals is key to China's low-carbon industrial transition. This paper examines both industrial development and technological innovations independently before offering conclusions.

Chinese industries account for a significant portion of the country's GHG emissions, largely due to their highly energy-intensive nature. China is a leading global exporter, and Western consumption patterns have contributed to the rapid growth in the country's carbon emissions. Several studies have analyzed the changes in $CO_2$ emissions and identified key determinants, including heat and electricity carbon emission coefficients, energy intensity, industrial structural shifts, industrial activity, and final fuel shifts [78–81]. Among the 36 industries studied, raw chemical or chemical material products, nonmetal mineral products, and smelting or pressing of ferrous metals are the most energy-intensive and deserve top priorities for enhancing energy efficiency. Researchers have found industrial output as the primary contributor to $CO_2$ emissions, with the level of industrial activity as the largest contributor to the industrial sector's $CO_2$ emissions. Energy intensity is the most significant factor affecting emission reduction. The specific $CO_2$ emission determinants in petrochemicals, food, iron and steel, and other industries have also been studied extensively [81–83].

These findings indicate that industrial output is the key factor in the absence of substantial decoupling between industrial production and GHG emissions. Li et al. tested the decoupling relationship between $CO_2$ emissions and economic growth in China and found only weak decoupling [79]. Similarly, Zhang and Da [84] and Zhao et al. [85] found weak decoupling between economic growth and carbon emissions during their studied periods. These results suggest that the economy has continued to grow with increased carbon emissions, underscoring the challenges of decarbonizing industrial production.

The iron and steel industry is a major contributor to China's energy consumption and $CO_2$ emissions (primarily from coal combustion), and increasing energy demand places significant pressure on GHG emission reduction goals. Sun et al. identified four factors affecting $CO_2$ emissions over time (emission coefficient, energy structure, energy consumption, and steel production output) [86], finding that output was the leading cause of $CO_2$ emissions. Lin and Wang comprehensively analyzed industrial $CO_2$ emission efficiency and mitigation potential, urging greater efforts to adopt energy-saving techniques and increase cooperation [87]. However, long-term low-carbon development requires energy substitution. Building construction is also a significant energy-demand sector, accounting for over 20% of China's final energy consumption and possessing significant potential for shaping energy perspectives for the future [88]. Thus, reducing GHG emissions associated with building construction is crucial for advancing low-carbon economic and urban development.

Studies have compared energy efficiency across various Chinese industries. Xia et al. comprehensively assessed the energy efficiency of 38 industries in China and identified the smelting and pressing of ferrous metals [89], the manufacture of raw chemical materials and chemical products, and the manufacture of non-metallic mineral products as having the lowest energy efficiency. Wu et al. established energy efficiency performance indexes for China's industrial sector based on $CO_2$ emissions and found that the greatest potential for improvement comes from technological advancements [90].

**4.1.3 Transition of the energy system.** Encouraging low-carbon development in cities and industrial sectors is vital, but observing China's overall energy demand and system structure can offer a more comprehensive perspective. It is crucial to evaluate the future energy

demand and structure in China and their relation to carbon emissions and the transition to a low-carbon energy system [73, 91].

Scenario analysis is a common method to forecast energy and typically involves low-carbon scenarios, implementation of low-carbon policies, and technological advancements. Forecast studies have been conducted on China's primary energy consumption trends, including those by Chai and Zhang [92] and Zhou et al. [93], as well as predictions of energy consumption, $CO_2$ emissions, and carbon intensity until 2030 or 2050 [94–96]. Research suggested that economic structure adjustments through improved efficiency could decrease carbon emissions growth rate in the long run. After 2030, GHG emissions will decrease in absolute terms through population stabilization and optimal industrial and energy structures, possibly hastened by low-carbon technologies like CCS. However, assumptions about short-term low-carbon changes vary, as projections based on adjusted energy and economic structures show limited reductions in total $CO_2$ emissions [97]. In the meantime, other studies argued the short-term difficulty in changing China's energy-intensive economic structure and limited capacity for energy efficiency improvement [98]. Thus, promoting non-fossil or low-carbon energy sources such as hydro, wind, and nuclear power is necessary for emission reduction targets, with renewable energy showing significant mitigation potential over energy efficiency and CCS [99]. A non-fossil energy strategy could also bolster China's energy security by reducing reliance on oil imports [100, 101]. This paper further examines two main issues: changes in energy demand and the transition of China's energy system towards non-fossil energy sources.

Several authors analyzed how overall Chinese energy demand and structure relate to carbon emissions. Considering the large annual $CO_2$ emissions and increases as well as a high emissions factor due to China's coal-dominated energy structure, He et al. assessed GHG emissions from fossil energy combustion [97]. Lin and Ouyang compared energy demand characteristics in China to those in the US, identifying economic growth, urbanization, and industrialization as significant contributors to demand increases, while improved energy intensity, technology, and higher energy prices lead to decreased consumption [102]. However, China's energy system relies heavily on high-carbon fossil fuels, particularly coal, which causes land, water, and air pollution and threatens environmental sustainability [92].

Researching low-carbon transitions and reducing GHG emissions in energy systems, particularly China's electricity system, is crucial. As the world's largest source of $CO_2$ emissions, the predominance of coal in China's electricity system is apparent [73]. Coal-fired power accounts for roughly 80% of total electricity generation and 40% of total emissions in China [103, 104], and shifting to low-carbon electricity is crucial for global climate change efforts [105].

Several studies provided insight into renewable energy development in China by analyzing the characteristics of the country's energy structure and exploring current issues. Jiang et al. examined policy measures, pressures, and challenges for hydro, wind, and nuclear power development and their potential contributions to GHG emission reduction and energy security [106]. Zhou et al. [107], Hu et al. [95], and Ren and Sovacool [108, 109] focused on the potential and contributions of non-fossil and low-carbon energy to carbon intensity reduction goals, energy security, and relative performance of hydropower, wind, solar, biomass, and nuclear energy. Lastly, Liu et al. analyzed the development status of renewable energy and its mitigation potential and found hydropower and wind had advantages in terms of technology maturity and price [99]. These studies can inform climate mitigation policies and low-carbon energy strategies.

## 4.2 Future directions for carbon reduction in China

In summary, after these general and cross-cutting comments, we will, as our last point, summarize our main suggestions for the three topics this paper focuses on: low-carbon cities, industry, and transition of the energy system.

Regarding low-carbon cities, conducting continuous inventory research for a city and observing emission changes over the long term. Assessing low-carbon practices in cities, particularly in China's low-carbon city pilot projects, to determine how low-carbon development goals are implemented. Examining the role of governments and governance issues during low-carbon city construction. Conducting a thorough social science analysis of local governance and urban planning, moving beyond formal mathematical modeling as found in current literature, particularly in English journals.

In terms of industry, examining the role of governments, enterprises, and citizens in developing low-carbon industries. Evaluating the relevance of management methods and public environmental awareness as drivers of low-carbon industry development. Conducting quantitative sector analyses, including multi-sector case studies that take a micro perspective to examine low-carbon technology usage, effects, and organizational features that promote or hinder implementation. Conducting cost analyses of low-carbon production processes for various industry sectors. Developing a roadmap with specific steps for improving the industrial structure and assessing the benefits and potential losses resulting from such changes.

Regarding transition of the energy system, identifying the drivers and hindering factors affecting the diffusion of renewable energy in China. Conducting detailed discussions on technology management strategies, including selection, system updates, and cost estimates, to further promote low-carbon production. Investigating collaboration mechanisms between governments, enterprises, and research institutes to facilitate low-carbon technology development. Analyzing the long-term costs and risks of nuclear energy, including nuclear waste management, for a balanced assessment of non-fossil options for China's energy mix.

In addition, the evaluation of $CO_2$ emissions in China still faces significant uncertainties. Future studies should address the following to improve carbon emissions inventory: 1) Considering coal qualities, such as ash content, in evaluating $CO_2$ emissions. Evaluating the different coal emission factors and their impact on China's $CO_2$ emissions. 2) Utilizing satellite data for carbon flux to improve existing $CO_2$ emission inventories regarding the amount, distribution, and variation. 3) Studying the methods and impacts of consumption-based emissions allocation to reduce emission responsibilities among regions.

## 4.3 Limitations

This study is not free of limitations. The bibliometric analysis data were collected from the WoS Core Collection database, considered the most suitable database for scientometrics research. However, the retrieved publications may have defects in terms of timeliness and completeness. In future studies, databases such as Scopus and Google Scholar may be included for comparison to obtain more comprehensive results. Another important limitation of scientometrics research is citation bias because the sole purpose of a citation is to emphasize the impact of the article and may not truly reflect its quality or relevance, thus underutilizing the available data. Different parameters can also lead to citation bias, such as the total number of publications (TP), total citations (TC), H index, and percentage centrality of CiteSpace. Finally, the different expressions of the same keywords may also lead to different search results and clustering results. This study includes synonyms in the search keywords as much as possible.

## 5. Conclusions

In this study, we conducted bibliometrics analysis on 1570 papers published between 2001 and 2021 from the WoS Core Collection database using Derwent Data Analyzer, CiteSpace, and Gephi. We demonstrated the status of research on China's carbon emissions from the annual publication volume, core research institution, institutional cooperation network, and high-yield journal aspects. To sum up, research on China's carbon emissions has developed rapidly in recent years. Due to the increasing importance of and attention to research on China's carbon emissions, future research should be enriched and deepened. In addition to national universities, more research institutions, including enterprises, local universities, think tanks, and social institutions, need to jointly promote relevant research in the future. Moreover, the research field needs expansion and enrichment, and frontier research across fields needs attention. The research hotspots were analyzed based on highly cited papers, keyword co-occurrence networks, high-frequency keywords, and burst detection. In conclusion, the field of China's carbon emissions shows the characteristics of multi-angle attention, multi-disciplinary research, multiple achievements, and multi-topic exploration. From different angles, scholars in different disciplines and fields have conducted in-depth research on the background, policy, market, and behavior, as well as case studies and comparative studies, contributing significantly to China's carbon emission reduction. The influencing factors, empirical analyses, and countermeasures of China's carbon emissions will still be the focus and hotspots of research. In the future, relevant research on technologies, industries, policies, effects, responses, experience references, comparative analyses, and model applications needs strengthening. In addition, research cooperation, especially domestic and international cooperation involving personnel, institutions, and issues, must be strengthened.

We extracted effective information and data from the existing literature related to China's carbon emissions, analyzed the historical context and current development, and explored the relevant research hotspots and prominent problems. On that basis, we made suggestions and provided an outlook for future research. This paper provides a reference for academic research to promote the development and internationalization of China's carbon emission research. In practice, it facilitates enterprises to adjust and establish the direction of technology research and development and provides governments with relevant decision-making references.

## Acknowledgments

This work was guided by the Environmental Protection Industry Association of Jiujiang City, Jiangxi Province. We thank Prof. Xianchuan Xie (Nanchang University) for constructive comments on manuscript revision. We thank the editors and reviewers for their valuable feedback on the manuscript.

## Author Contributions

**Conceptualization:** Wei Liu.

**Data curation:** Caiyun Chen, Wei Liu.

**Formal analysis:** Caiyun Chen, Wei Liu.

**Funding acquisition:** Caiyun Chen, Wei Liu.

**Methodology:** Wei Liu.

**Project administration:** Wei Liu.

**Resources:** Wei Liu.

**Supervision:** Wei Liu.

**Validation:** Wei Liu.

**Writing – original draft:** Caiyun Chen.

**Writing – review & editing:** Caiyun Chen.

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
