## [Decision Letter · Decision Letter 0]

8 Jun 2023

PONE-D-23-15921Analyzing China's Efforts and Future Trends in Carbon Emission Reduction from the Perspective of BibliometricsPLOS ONE

Dear Dr. Liu,

Thank you for submitting your manuscript to PLOS ONE. After careful consideration, we feel that it has merit but does not fully meet PLOS ONE’s publication criteria as it currently stands. Therefore, we invite you to submit a revised version of the manuscript that addresses the points raised during the review process.

We look forward to receiving your revised manuscript.

Kind regards,

Rita Yi Man Li

Academic Editor

PLOS ONE

Journal Requirements:

   "This work was supported by Jiujiang traditional petrochemical industry green low carbon Upgrading Research Project entrusted by Jiujiang Environmental 

Protection Industry Association (HX202205130002)."

   "NO"

Additional Editor Comments:

Comments for Analyzing China's Efforts and Future Trends in Carbon Emission Reduction from the Perspective of Bibliometrics

1.How can we Analyzing China's Efforts from the Perspective of Bibliometrics?

Remove the location from title

1.the authors may need to reduce the overall length of the abstract part, the relative background may be simplified, and the research methods could be briefly mentioned.

2.“Since the pre-industrial era, increasing global warming has led to melting polar glaciers, hurricane damage, accelerated species extinction, rising sea levels, and changing rainfall patterns”. Please add reference to support this opinion.

3.“the global average temperature may rise by 1.5 by 2030-2052.” Where does this result come from? The authors may add a reference here.

4.some of the reference in the introduction part may seem a bit outdated, the authors may consider revising them.

5.Why almost all the CO2 emission data and papers are before 2010? Please add research after 2010.

https://www.frontiersin.org/articles/10.3389/fenvs.2022.949959/full?utm_source=dlvr.it&utm_medium=twitter

6.“China's carbon emissions have increasingly become a hot issue in the international community. Relevant research has been continuously enriched and developed. Many research results were published in journal articles. The research contents and perspectives are rich,” why this sentence stays here? This sentence could be simplified.

7.The authors should shorten length of the introduction part.

8.There should be a brief introduction of the coming sections of this paper at the end of the introduction part.

9.There should be a section with title “Literature review” which discuss the research background and current gaps, and also this section may be helpful regarding controlling the length of the introduction part.

10.The title Merhodology may be changed to Methods instead.

11.The authors may take a discussion about reasons for the selection of WoSviewer, Gephi, Citespace in this research and where have these tools been used before:

https://www.sciencedirect.com/science/article/abs/pii/S0925753522001291

https://www.mdpi.com/2071-1050/14/1/562

12.Reference about the knowledge map is necessary in the research methods part.

13.A descriptive statistics table should be added, which may contains the published year, journal name., etc.

14.There should be a discussion section.

15.Figure 5 is unclear, please enlarge.

16.Burst words are keywords that suddenly increase in number within a certain…please add citation.

17.Lines 624-654 should avoid numbering, please use paragraph.

Reviewers' comments:

Reviewer's Responses to Questions

**Comments to the Author**

1. Is the manuscript technically sound, and do the data support the conclusions?

Reviewer #1: Yes

Reviewer #2: Yes

2. Has the statistical analysis been performed appropriately and rigorously? 

Reviewer #1: Yes

Reviewer #2: Yes

3. Have the authors made all data underlying the findings in their manuscript fully available?

Reviewer #1: Yes

Reviewer #2: Yes

4. Is the manuscript presented in an intelligible fashion and written in standard English?

Reviewer #1: Yes

Reviewer #2: No

5. Review Comments to the Author

Reviewer #1: The research method is appropriate, the data analysis is more systematic, the conclusion is logical, the picture is clear, and the reference is standard.The article is generally good enough to be published.

Reviewer #2: The logical structure of the article is clear, while the topic and content meet the requirements of originality. Language expressions and methods used in the data analysis section are detailed, rigorous, and standardized. Conclusions are presented in an appropriate manner and supported by data. I propose the following suggestions for authors to refer to.

The article has some grammar issues that require further improvement and correction by the author.

The references are relatively outdated, and the author can appropriately increase the proportion of references cited in the past five years.

Especially, it is necessary to add professional journal papers related to bibliometrics, as follows.

Wang, X. (2022). Research on the discourse power evaluation of academic journals from the perspective of multiple fusion: taking Medicine, General and Internal journals as an example. Journal of Information Science, 01655515221107334.

Wang, X. (2022), "Characteristics analysis and evaluation of discourse leading for academic journals: perspectives from multiple integration of altmetrics indicators and evaluation methods", Library Hi Tech, Vol. ahead-of-print No. ahead-of-print. https://doi.org/10.1108/LHT-04-2022-0195

Wang, X. and Feng, X. (2022), "Research on the relationships between discourse leading indicators and citations: perspectives from altmetrics indicators of international multidisciplinary academic journals", Library Hi Tech, Vol. ahead-of-print No. ahead-of-print. https://doi.org/10.1108/LHT-09-2021-0296

Wang, X., Feng, X. and Guo, K. (2023), "Research hotspots and prospects of ethics education of science and technology in China based on bibliometrics", Library Hi Tech, Vol. 41 No. 2, pp. 454-473. https://doi.org/10.1108/LHT-06-2022-0298

6. PLOS authors have the option to publish the peer review history of their article (what does this mean?). If published, this will include your full peer review and any attached files.

Reviewer #1: No

Reviewer #2: No

---

## [Author Response · Author response to Decision Letter 0]

29 Jun 2023

Dear editor and reviewers：

Thank you very much for your email dated June 8, 2023, informing us the Referees’ comments on our manuscript entitled “Analyzing China's Efforts and Future Trends in Carbon Emission Reduction from the Perspective of Bibliometrics” (PONE-D-23-15921). We appreciate all of your time and work in processing our manuscript. Your insightful comments and suggestions are valuable and very helpful for further revising and improving our paper.

After receiving the email, we read the comments and suggestions of the editor and reviewers with great interest. We revised our manuscript point by point based on their comments and suggestions and provided a detailed point-by-point response, as described below. Reviewers' comments are listed in black italics and specific questions have been numbered. Our responses are given in regular black font, and relevant text is highlighted in red and blue in the revised manuscript.

We hope our revisions will meet your satisfaction.

Once again, thank you for your help. We are looking forward to your final decision at your earliest convenience.

Sincerely yours,

Wei Liu

Replies to reviewer’s comments：

1.How can we Analyzing China's Efforts from the Perspective of Bibliometrics? Remove the location from title.

Response:

Many thanks to the reviewers for their suggestions and reminders, which we think will add professionalism and rigor to the manuscript. As suggested by the reviewer, we have revised the title to make it more relevant. Please see the revised manuscript for details (lines 1-5).

2.The authors may need to reduce the overall length of the abstract part, the relative background may be simplified, and the research methods could be briefly mentioned.

Response:

Thank you very much for your suggestion. We have made revisions to the abstract as requested. We have simplified the background introduction and emphasized the explanation of the research methodology and the results. Please see the revised manuscript for details (lines 17-47).

3.“Since the pre-industrial era, increasing global warming has led to melting polar glaciers, hurricane damage, accelerated species extinction, rising sea levels, and changing rainfall patterns”. Please add reference to support this opinion.

Response:

Thank you very much for your suggestion. We have added relevant literature to make the description of the research background more reliable. Furthermore, we made detailed modifications to the structure and content of the introduction. Please see the revised manuscript for details (revised introduction, lines 50-141).

References:

[1] B. Marzeion, G. Kaser, F. Maussion, N. Champollion, Limited influence of climate change mitigation on short-term glacier mass loss, Nature Climate Change, 8 (2018) 305-308.

[2] C. Taylor, T.R. Robinson, S. Dunning, J. Rachel Carr, M. Westoby, Glacial lake outburst floods threaten millions globally, Nature Communications, 14 (2023) 487.

4.“the global average temperature may rise by 1.5 by 2030-2052.” Where does this result come from? The authors may add a reference here.

Response:

Thank you very much for your suggestion. We have added relevant literature to increase the reliability and scientific rigor of this section.

References:

[1] IPCC, Climate Change 2022: Impacts, Adaptation and Vulnerability:Summary for Policymakers, 2022.

[2] R. Biesbroek, S.J. Wright, S.K. Eguren, A. Bonotto, I.N. Athanasiadis, Policy attention to climate change impacts, adaptation and vulnerability: a global assessment of National Communications (1994–2019), Climate Policy, 22 (2022) 97-111.

5.Some of the reference in the introduction part may seem a bit outdated, the authors may consider revising them.

Response:

Many thanks to the reviewers for their suggestions and reminders, which we think will add professionalism and rigor to the manuscript and make it more convincing. We have updated the research background with the latest literature on relevant studies in recent years, which will enhance the reliability and scientific rigor of our work.

References:

[1] R. Biesbroek, S.J. Wright, S.K. Eguren, A. Bonotto, I.N. Athanasiadis, Policy attention to climate change impacts, adaptation and vulnerability: a global assessment of National Communications (1994–2019), Climate Policy, 22 (2022) 97-111.

[2] C. Taylor, T.R. Robinson, S. Dunning, J. Rachel Carr, M. Westoby, Glacial lake outburst floods threaten millions globally, Nature Communications, 14 (2023) 487.

[3] R. Biesbroek, S.J. Wright, S.K. Eguren, A. Bonotto, I.N. Athanasiadis, Policy attention to climate change impacts, adaptation and vulnerability: a global assessment of National Communications (1994–2019), Climate Policy, 22 (2022) 97-111.

[4] R.Y.M. Li, Q. Wang, L. Zeng, H. Chen, A Study on Public Perceptions of Carbon Neutrality in China: has the Idea of ESG Been Encompassed?, Frontiers in Environmental Science, 10 (2023).

[5] R. Sun, K. Wang, X. Wang, J. Zhang, China's Carbon Emission Trading Scheme and Firm Performance, (2022).

[6] X. Wang, Research on the discourse power evaluation of academic journals from the perspective of multiple fusion: Taking Medicine, General and Internal journals as an example, Journal of Information Science, (2022) 01655515221107334.

[7] X. Wang, Characteristics analysis and evaluation of discourse leading for academic journals: perspectives from multiple integration of altmetrics indicators and evaluation methods, Library Hi Tech, ahead-of-print (2022).

[8] X. Wang, X. Feng, Research on the relationships between discourse leading indicators and citations: perspectives from altmetrics indicators of international multidisciplinary academic journals, Library Hi Tech, ahead-of-print (2022).

[9] L. Wang, Y. Xu, T. Qin, M. Wu, Z. Chen, Y. Zhang, W. Liu, X. Xie, Global trends in the research and development of medical/pharmaceutical wastewater treatment over the half-century, Chemosphere, 331 (2023) 138775.

[10] L. Zeng, R.Y.M. Li, Construction safety and health hazard awareness in Web of Science and Weibo between 1991 and 2021, Safety Science, 152 (2022) 105790.

[11] L. Zeng, R.Y. Li, J. Nuttapong, J. Sun, Y. Mao, Economic Development and Mountain Tourism Research from 2010 to 2020: Bibliometric Analysis and Science Mapping Approach, in: Sustainability, 2022.

[12] X. Wang, X. Feng, K. Guo, Research hotspots and prospects of ethics education of science and technology in China based on bibliometrics, Library Hi Tech, 41 (2023) 454-473.

6.Why almost all the CO2 emission data and papers are before 2010? Please add research after 2010.

https://www.frontiersin.org/articles/10.3389/fenvs.2022.949959/full?utm_source=dlvr.it&utm_medium=twitter

Response:

Thank you very much for the reminder, this is a very crucial addition. We have included the latest research data and streamlined some content in this section to control the length of the introduction. As the reviewer mentioned, this is a recent publication that discusses the latest hot topics in China's carbon emissions research. We have added this reference and read it carefully, and we are inspired by carefully reading the literature of Li et al. (2023). Please see the revised manuscript for details (revised introduction, lines 50-141).

References:

[1] R.Y.M. Li, Q. Wang, L. Zeng, H. Chen, A Study on Public Perceptions of Carbon Neutrality in China: has the Idea of ESG Been Encompassed?, Frontiers in Environmental Science, 10 (2023).

7.“China's carbon emissions have increasingly become a hot issue in the international community. Relevant research has been continuously enriched and developed. Many research results were published in journal articles. The research contents and perspectives are rich,” why this sentence stays here? This sentence could be simplified.

Response:

Thank you very much for your suggestion. As you mentioned, the information provided in this paragraph was limited and the expression was not concise enough. We simplified this content and reviewed the entire manuscript to ensure that the language is clear and easy to understand. please see the revised manuscript for details (lines 105-107).

8 .The authors should shorten length of the introduction part.

Response:

Thank you very much for your suggestion. As you mentioned, the introduction section was too lengthy, so as per the reviewer's request, we have made detailed revisions to the introduction with the objectives of (1) controlling the length of the introduction, and (2) improving the logical coherence of the text, thus enhancing the rigor and readability of the paper. Additionally, we have added the latest relevant research literature as requested by the reviewer, which greatly improved the quality of the introduction. Please see the revised manuscript for details (revised introduction, lines 50-141).

9. There should be a brief introduction of the coming sections of this paper at the end of the introduction part.

Response:

Many thanks to the reviewers for their suggestions and reminders, which we think will add professionalism and rigor to the manuscript and make it more convincing. As suggested by the reviewer, in the last paragraph, we described the purpose of conducting this study and the research contents of the upcoming chapters. Please see the revised manuscript for details (lines 119-141).

10.There should be a section with title “Literature review” which discuss the research background and current gaps, and also this section may be helpful regarding controlling the length of the introduction part.

Response:

Many thanks to the reviewers for their suggestions and reminders, which we think will add professionalism and rigor to the manuscript and make it more convincing. As suggested by the reviewer, we have made detailed revisions to the introduction with the objectives of (1) controlling the length of the introduction, and (2) improving the logical coherence of the text, thus enhancing the rigor and readability of the paper. As requested by the reviewer, we introduced the latest reports on relevant research in the second and third paragraph of the introduction. Although we did not add a separate “literature review” section, we briefly summarized the latest literature on the research background and methodology , which also helped to control the overall length of the manuscript. We appreciate the reviewer's professional feedback, which greatly improved the quality of the introduction, especially with the addition of many new relevant references. Please see the revised manuscript for details (revised introduction, lines 50-141).

11.The title Merhodology may be changed to Methods instead.

Response:

Many thanks to the reviewers for their suggestions and reminders. As suggested by the reviewer, we have revised the title to make it more relevant. Please see the revised manuscript for details (lines 159, 160, 175).

12.The authors may take a discussion about reasons for the selection of WoSviewer, Gephi, Citespace in this research and where have these tools been used before: https://www.sciencedirect.com/science/article/abs/pii/S0925753522001291

https://www.mdpi.com/2071-1050/14/1/562

Response: 

Please accept my sincere thanks for your professional comments and suggestions. As the reviewer mentioned, Analysis tools are crucial and are not described in sufficient detail in our research methodology. This is a very important list of references, which we have read carefully and added relevant and cited in this article.

References:

[1] L. Zeng, R.Y.M. Li, Construction safety and health hazard awareness in Web of Science and Weibo between 1991 and 2021, Safety Science, 152 (2022) 105790.

[2] L. Zeng, R.Y. Li, J. Nuttapong, J. Sun, Y. Mao, Economic Development and Mountain Tourism Research from 2010 to 2020: Bibliometric Analysis and Science Mapping Approach, in: Sustainability, 2022.

13.Reference about the knowledge map is necessary in the research methods part.

Response:

Thank you very much for your suggestion. As you mentioned, We have referred to and cited relevant research on knowledge maps in the research methods part of our study.

References:

[1] L. Wang, Y. Xu, T. Qin, M. Wu, Z. Chen, Y. Zhang, W. Liu, X. Xie, Global trends in the research and development of medical/pharmaceutical wastewater treatment over the half-century, Chemosphere, 331 (2023) 138775.

[2] L. Zeng, R.Y.M. Li, Construction safety and health hazard awareness in Web of Science and Weibo between 1991 and 2021, Safety Science, 152 (2022) 105790.

[3] L. Zeng, R.Y. Li, J. Nuttapong, J. Sun, Y. Mao, Economic Development and Mountain Tourism Research from 2010 to 2020: Bibliometric Analysis and Science Mapping Approach, in: Sustainability, 2022.

14. A descriptive statistics table should be added, which may contains the published year, journal name., etc.

Response:

Thank you very much for your suggestion. As you mentioned, We added descriptive statistics on relevant research information, including the number and proportion of journal publications, highly cited articles, and publication years. Please see the revised manuscript for details (Tables 1 and 2, lines 332, 368).

15.There should be a discussion section.

Response:

Many thanks to the reviewers for their suggestions and reminders. As you mentioned, the discussion section is necessary and crucial, so in chapter three, we conducted a thorough analysis of the results obtained through econometrics and statistical analysis in each subsection. Furthermore, we adjusted the structure of sections 3.3 and 4, adding a discussion section. We believe this adjustment is a professional recommendation that makes the article structure more rigorous and the discussion logic clearer. Once again, we appreciate the valuable suggestions from the reviewers, we will continue to maintain this approach in our future writing. Please see the revised manuscript for details (4. Discussion, line 460).

16.Figure 5 is unclear, please enlarge.

Response:

Thank you very much for your suggestion. As you mentioned, we enlarged Figure 5 to make it clearer. Please see the revised manuscript for details (Figure 5, line 394).

17.Burst words are keywords that suddenly increase in number within a certain…please add citation.

Response:

Thank you very much for your suggestion. As you mentioned, burst words can present research hotspots or trends during a certain period, therefore literature related to key nodes is worth referencing. We added some literature related to this study to explain the relevant information of the burst words.

References:

[1] J. Chen, W. Gui, Y. Huang, The impact of the establishment of carbon emission trade exchange on carbon emission efficiency, Environmental Science and Pollution Research, 30 (2023) 19845-19859.

[2] S. Wang, Y. Yu, T. Jiang, J. Nie, Analysis on carbon emissions efficiency differences and optimization evolution of China’s industrial system: An input-output analysis, PLOS ONE, 17 (2022) e0258147.

[3] Y. Yu, J. Su, Y. Du, Impact of global value chain and technological innovation on China’s industrial greenhouse gas emissions and trend prediction, International Journal of Environmental Science and Technology, (2023).

[4] Z. Zuo, H. Guo, J. Cheng, An LSTM-STRIPAT model analysis of China’s 2030 CO2 emissions peak, Carbon Management, 11 (2020) 577-592.

[5] X. Yu, C. Tan, China’s process-related greenhouse gas emission dataset 1990–2020, Scientific Data, 10 (2023) 55.

[6] J. Han, W. Zhu, C. Chen, Identifying Emissions Reduction Opportunities in International Bilateral Emissions Trading Systems to Achieve China’s Energy Sector NDCs, in: International Journal of Environmental Research and Public Health, 2023.

[7] J. Cao, M. Ho, Q. Liu, Analyzing multi-greenhouse gas mitigation of China using a general equilibrium model, Environmental Research Letters, 18 (2023) 025001.

[8] Q. Zhang, B. Gu, H. Zhang, Q. Ji, Emission reduction mode of China's provincial transportation sector: Based on “Energy+” carbon efficiency evaluation, Energy Policy, 177 (2023) 113556.

[9] C. Rao, Q. Huang, L. Chen, M. Goh, Z. Hu, Forecasting the carbon emissions in Hubei Province under the background of carbon neutrality: a novel STIRPAT extended model with ridge regression and scenario analysis, Environmental Science and Pollution Research, 30 (2023) 57460-57480.

[10] G. Pang, Z. Ding, X. Shen, Spillover effect of energy intensity reduction targets on carbon emissions in China, Frontiers in Environmental Science, 11 (2023).

[11] C. Xu, F. Liu, Y. Zhou, R. Dou, X. Feng, B. Shen, Manufacturers' emission reduction investment strategy under carbon cap-and-trade policy and uncertain low-carbon preferences, Industrial Management & Data Systems, ahead-of-print (2023).

[12] Q.-H. Zeng, L.-Y. He, Study on the synergistic effect of air pollution prevention and carbon emission reduction in the context of "dual carbon": Evidence from China's transport sector, Energy Policy, 173 (2023) 113370.

18.Lines 624-654 should avoid numbering, please use paragraph.

Response:

Thank you very much for the reminder, we have made the change, Please see the revised manuscript for details (lines 685-715).

Review Comments to the Author:

19.The article has some grammar issues that require further improvement and correction by the author.

Response: 

Thank you very much for your suggestion. We apologize for any unclear language in our manuscript, and we take the reviewer's comments very seriously. We have taken steps to address this issue by seeking help from professional organizations and editors to review and revise the manuscript. Additionally, we have provided evidence of the language editing work done.

20.The references are relatively outdated, and the author can appropriately increase the proportion of references cited in the past five years.

Especially, it is necessary to add professional journal papers related to bibliometrics, as follows.

Response: 

Please accept my sincere thanks for your professional comments and suggestions. This is a very important list of references, which we have read carefully and added relevant and important elements to this article. As mentioned above, these review comments not only enhance the quality of this review article but also deepens the authors' understanding of bibliometrics.

References:

[1] Wang, X. (2022). Research on the discourse power evaluation of academic journals from the perspective of multiple fusion: taking Medicine, General and Internal journals as an example. Journal of Information Science, 01655515221107334.

[2]Wang, X. (2022), "Characteristics analysis and evaluation of discourse leading for academic journals: perspectives from multiple integration of altmetrics indicators and evaluation methods", Library Hi Tech, Vol. ahead-of-print No. ahead-of-print. https://doi.org/10.1108/LHT-04-2022-0195

[3] Wang, X. and Feng, X. (2022), "Research on the relationships between discourse leading indicators and citations: perspectives from altmetrics indicators of international multidisciplinary academic journals", Library Hi Tech, Vol. ahead-of-print No. ahead-of-print. https://doi.org/10.1108/LHT-09-2021-0296

[4] Wang, X., Feng, X. and Guo, K. (2023), "Research hotspots and prospects of ethics education of science and technology in China based on bibliometrics", Library Hi Tech, Vol. 41 No. 2, pp. 454-473. https://doi.org/10.1108/LHT-06-2022-0298

---

## [Editor Report · Decision Letter 1]

2 Jul 2023

Advances and Future Trends in Research on Carbon Emissions Reduction in China from the Perspective of Bibliometrics

PONE-D-23-15921R1

Dear Dr. Liu,

We’re pleased to inform you that your manuscript has been judged scientifically suitable for publication and will be formally accepted for publication once it meets all outstanding technical requirements.

Kind regards,

Rita Yi Man Li

Academic Editor

PLOS ONE

Additional Editor Comments (optional):

4.2, why China only?

4.2, shorten this section, combines some point forms to paragraphs.

Figure 2 needs higher resolution.

References, DESA. The sustainable development goals report 2018. 2018. UNEP. Emissions gap report 2021: The Heat Is On. 2022. Avoid using references that are not journal articles like these.

Replace conference proceeding by journal articles.
---

## [Editor Report · Acceptance letter]

12 Jul 2023

PONE-D-23-15921R1 

Advances and Future Trends in Research on Carbon Emissions Reduction in China from the Perspective of Bibliometrics 

Dear Dr. Liu:

I'm pleased to inform you that your manuscript has been deemed suitable for publication in PLOS ONE. Congratulations! Your manuscript is now with our production department. 

Kind regards, 

on behalf of

Dr. Rita Yi Man Li 

Academic Editor

PLOS ONE